# A rigid and healable polymer cross-linked by weak but abundant Zn(II)-carboxylate interactions

Jian-Cheng Lai[1], Lan Li[2,3], Da-Peng Wang[1], Min-Hao Zhang[1], Sheng-Ran Mo[1], Xue Wang[1], Ke-Yu Zeng[1], Cheng-Hui Li[1,3], Qing Jiang[2,3], Xiao-Zeng You[1] & Jing-Lin Zuo [1]

Achieving a desirable combination of solid-like properties and fast self-healing is a great challenge due to slow diffusion dynamics. In this work, we describe a design concept that utilizes weak but abundant coordination bonds to achieve this objective. The designed PDMS polymer, crosslinked by abundant Zn(II)-carboxylate interactions, is very strong and rigid at room temperature. As the coordination equilibrium is sensitive to temperature, the mechanical strength of this polymer rapidly and reversibly changes upon heating or cooling. The soft–rigid switching ability σ, defined as $G'_{max}/G'_{min}$, can reach 8000 when $\Delta T = 100\,°C$. Based on these features, this polymer not only exhibits fast thermal-healing properties, but is also advantageous for various applications such as in orthopedic immobilization, conductive composites/adhesives, and 3D printing.

[1] State Key Laboratory of Coordination Chemistry, School of Chemistry and Chemical Engineering, Nanjing National Laboratory of Microstructures, Collaborative Innovation Center of Advanced Microstructures, Nanjing University, Nanjing 210093, P. R. China. [2] Department of Sports Medicine and Adult Reconstructive Surgery, Drum Tower Hospital affiliated to Medical School of Nanjing University, Nanjing 210008, P. R. China. [3] Institute of Medical 3D Printing, Nanjing University, Nanjing 210093, P. R. China. Correspondence and requests for materials should be addressed to C.-H.L. (email: chli@nju.edu.cn) or to J.-L.Z. (email: zuojl@nju.edu.cn)

Self-healing abilities are an important survival feature in nature, as these properties allow living beings to repair damage when wounded. The self-healing abilities of living beings have inspired scientists to invent various methods for restoring the functionality of damaged materials[1–7]. To date, many synthetic polymers have been designed to self-heal by encapsulating healing agents (in microcapsules[8–10] or microvascular[11–13] networks) or incorporating dynamic covalent bonds (such as alkoxyamine[14–16], disulfide[17–21], boronic ester and boroxine[22–24] bonds or bonds formed by Diels-Alder reaction[25–27]) or non-covalent interactions (such as hydrogen bonds[28–30], π–π stacking interactions[31,32], host–guest interactions[33,34], ionic interactions[35–37] and metal-ligand interactions[38–43]) into the polymer matrix. However, for most self-healing materials, there is often a trade-off between the mechanical properties and the dynamic healing; strong bonds result in mechanically robust but less dynamic systems, precluding autonomous healing, while weak bonds afford dynamic healing, but yield relatively soft materials. Therefore, it is quite challenging to achieve self-healing in strong and solid-like materials[27,29,42].

Nature has given us hints to solve this conundrum. Hydrogen bonds are considerably weaker than other interactions, but these weak interactions can form very strong materials in some situations. For example, chitin, which consists of polysaccharides (sugars) assembled through extensive hydrogen bonding, has amazing high mechanical strength and serves as a protective shell (such as lobsters claws, beetle carapaces, and tree bark) for living organism[44,45]. As another example, at low temperature, water molecules aggregate into solid ice via ordered hydrogen bonding. The compressive strength of ice can be up to 5–25 MPa over the temperature range −10 °C to −20 °C[46]. Recently, Aida et al. reported a mechanically robust yet readily repairable polymer that was cross-linked by a dense hydrogen bonding network[47]. These phenomena indicate that weak bonds that are sufficiently abundant and arranged in an orderly manner, can lead to materials with excellent mechanical strength.

Inspired by nature, herein we describe a design concept that utilizes weak but abundant coordination bonds to achieve rigid and healable materials. The coordination bonds used in our study are weak but still significantly stronger than hydrogen bonds. Therefore, the resulting polymer is very strong (with flexural Young's modulus as high as 480 MPa) and rigid (with an elongation at break smaller than 4%) at room temperature. The coordination equilibrium is sensitive to temperature; thus, the mechanical strength of our polymer exhibits distinct (as high as almost 4 orders of magnitude in a narrow temperature range ($\Delta T < 100$ °C)), fast (within tens of seconds) and reversible change upon heating or cooling. Such features make our polymer applicable in various situations. For example, due to its rapid softening and hardening property, our polymer can be used in orthopedic immobilization to replace traditional plaster casting, and it also has the advantages of being lightweight, removable and recyclable. Our polymer can also be used for 3D printing since it turns into a viscous liquid upon heating to 120 °C and quickly forms a rigid solid upon cooling. With its thermal healing properties, objects made of our polymer using 3D printing can be healed when damaged. We can also obtain large or complex objects with only a small 3D printer by taking advantage of the healing processes of this material. Thus we can combine the advantages of modern 3D-printing processes and traditional brick-and-mortar operation using our materials. Moreover, our polymer can be used to prepare conductive composites/adhesives that are reshapable, healable, and 3D printable.

## Results

### Material design and characterizations

We selected Zn(II)-carboxylate interactions for our design. According to hard-soft-acid-base theory in coordination chemistry, Zn(II) is a borderline acid while carboxylate is a hard base. Therefore, the coordination bond between Zn(II) and carboxylate is relatively weak but still stronger than a hydrogen bond. A designed linear poly(dimethylsiloxane) (PDMS) polymer backbone, denoted **PDMS-COOH**, which possess abundant carboxylic acid groups (approximately one carboxylic acid for every two −O−Si−O− group) along the polymer backbone, was synthesized. The polymer **PDMS-COO-Zn** was synthesized from **PDMS-COOH** according to Fig. 1a and Supplementary Fig. 1. Detailed synthesis and characterization information can be found in the Supplementary Information (Supplementary Fig. 2–7). According to the isothermal calorimetric titrations (ITC) studies on **PDMS-COO-Zn** (Fig. 1b and Supplementary Table 1), the association constant ($K_a$) was $4.10 \times 10^4$ M$^{-1}$, which is quite small compared to common coordination bonds but is still much higher than that of hydrogen bond[48,49]. The crosslinked **PDMS-COO-Zn** network was obtained as a solid material after the reaction, and it can be grounded into a powder and hot pressed into a block sample (Fig. 1c). According to the FT-IR spectra (Supplementary Fig. 4) and the crystal structures of the Zn(II)-carboxylate complexes reported in the literature[50,51], the Zn(II)-carboxylate tetragonal coordination interactions are the dominant crosslinking sites within the polymer matrix (Fig. 1d). No apparent aggregation of the Zn(II)-carboxylate complexes was observed in the small-angle X-ray scattering (SAXS) analysis (Supplementary Fig. 5) and energy dispersive X-ray spectroscopy (EDS) data (Supplementary Fig. 6). Thermal gravimetric analysis (TGA) showed that the polymer was stable below 150 °C (Supplementary Fig. 7). Dissolution experiments revealed that the polymer had good solvent resistance to most solvents (Supplementary Fig. 8). The glass transition temperature ($T_g$) of the polymer was approximately 55.7 °C according to differential scanning calorimetry (DSC) (Supplementary Fig. 9). These features indicate that at room temperature, the polymer behaves like a thermoset polymer crosslinked by strong covalent bonds[52–54].

### Mechanical properties

The key mechanical properties determined from flexural testing are summarized in Supplementary Table 2. The material has a high flexural Young's modulus of $478.13 \pm 20.25$ MPa (mean ± s.d., $n = 4$) at a testing rate of 20 mm min$^{-1}$ at 25 °C (Fig. 2a), which is more than two times higher than that of the stiff and healable PDMS polymer cross-linked by strong covalent boroxine bonds[27]. Moreover, only slight elongation ( < 4% before fracturing) is observed upon applying a flexural strength of 9.16 MPa, which confirms the rigidity of the material. In fact, the samples are similar to ceramic materials and will shatter upon compression (Supplementary Movie 1). The enhanced strength of the material is attributed to the high density of the weak crosslinking coordination interactions between the carboxylate and the Zn(II) ions. Figure 2b shows that a thin plate (2.5 mm × 25 mm × 70 mm, 7.3 g) could withstand a heavy load of 200 g without obvious bending or breaking within 1 h in the single cantilever mode. These features indicate that our material is rigid and strong with very poor molecular segment mobility under ambient conditions.

The rigid polymer becomes soft and viscoelastic upon heating. When the temperature is higher than 50 °C, the material becomes bendable and does not break even under 50% strain (Fig. 2a). The temperature-dependent rheological measurements (Fig. 2c) show that the storage modulus is 470 MPa at 25 °C and 0.06 MPa at 125 °C. The soft–rigid switch ability, σ, defined as G'$_{max}$ /G'$_{min}$, can reach approximately 8000 when $\Delta T = 100$ °C (from 25 °C to 125 °C). Such a significant change in the mechanical strength under a narrow temperature range has never been reported for self-healing polymers[55,56]. Interestingly, the change in the

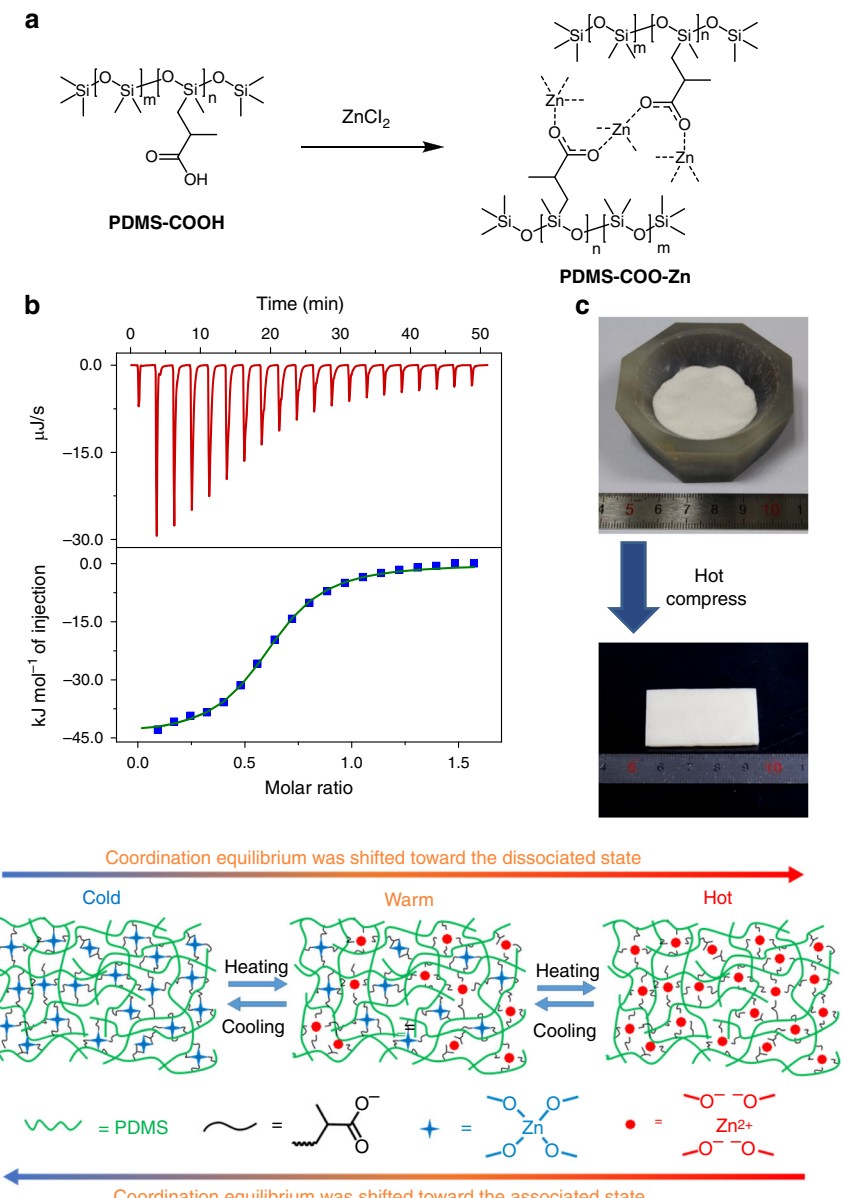

**Fig. 1** Preparation and characterizations of **PDMS-COO-Zn** polymer. **a** Synthesis and structure of **PDMS-COO-Zn** polymer. **b** The ITC titration data of the **PDMS-COO⁻** with ZnCl₂ in anhydrous ethanol at 25 °C. **c** Hot compressing of the powder lead to the block sample. **d** Schematic structure of the polymer network at different temperatures

mechanical strength upon heating is highly reversible. When cooled after heating, the soft and viscoelastic polymer quickly converts back into its strong and rigid state. As revealed by the cyclic rheological measurements (Fig. 2d), the temperature dependence of the modulus of the material during cooling is nearly the same as that during heating. The frequency sweep measurements at different temperatures show that the relaxation time of the polymer decreases with the increase of temperature (Supplementary Fig. 10). The temperature-dependent dynamic mechanical analysis (DMA) study shows that the elastic modulus dropped rapidly from 800 MPa at room temperature (25 °C) to 0.9 MPa at 80 °C (Fig. 2e). Owning to the association and dissociation of water molecules to Zn(II) centers, the **PDMS-COO-Zn** polymer also shows distinct and reversible changes in mechanical strength and bulk relaxation time at different relative humidity (Supplementary Fig. 11 and 12).

The temperature-dependence of the mechanical properties of the **PDMS-COO-Zn** polymer is illustrated in Fig. 1d. The short

PDMS linear chains are crosslinked by Zn(II)-carboxylate interactions into a three-dimensional network. The weak but abundant Zn(II)-carboxylate interactions endow the polymer with high mechanical strength. When heated, the $Zn^{2+}$ + PDMS-COO⁻ ↔ $Zn^{2+}$(⁻OOC-PDMS) equilibrium shifts toward the disassociated state, increasingly generating non-crosslinked **PDMS-COO⁻** chains, reducing the mechanical strength and enhancing the polymer chain mobility. Therefore, at higher temperatures, more non-crosslinked **PDMS-COO⁻** chains are generated, and consequently, the polymer becomes softer and more viscoelastic. Upon cooling, the $Zn^{2+}$ + PDMS-COO⁻ ↔ $Zn^{2+}$(⁻OOC-PDMS) equilibrium shifts toward the associated state, restoring the densely crosslinked three-dimensional network. On the other hand, as shown in Fig. 1f, the **PDMS-COO-Zn** polymer has high coefficients of thermal conductivity (1.0–1.5 W m⁻¹ K⁻¹) in the temperature range of 25–80 °C. Therefore, the mechanical properties of the **PDMS-COO-Zn** polymer are highly sensitive to temperature, which is beneficial for healing and

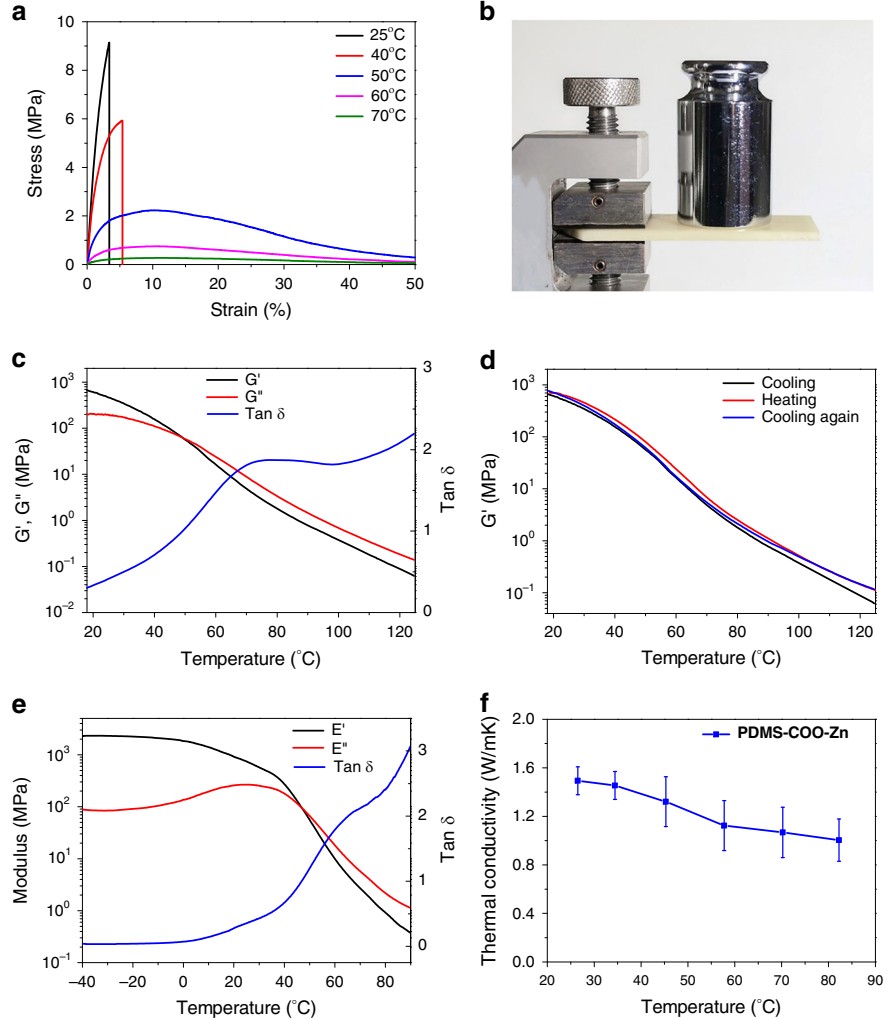

**Fig. 2** Mechanical and thermal properties of **PDMS-COO-Zn** polymer. **a** Three-point flexural stress-strain curves of **PDMS-COO-Zn** under various temperature. **b** The weight-bearing test of **PDMS-COO-Zn** polymer. **c** Temperature dependent rheology measurements of **PDMS-COO-Zn** polymer. **d** Cyclic temperature-sweep rheology measurements of **PDMS-COO-Zn** polymer (with the temperature changing rate of 2 °C min$^{-1}$). **e** Temperature-sweep dynamic mechanical analysis result of **PDMS-COO-Zn** polymer. **f** Thermal conductive properties of **PDMS-COO-Zn** polymer. Error bars are the s.d. from triplicate measurements

reshaping upon heating. The choice of metal ions and the metal-to-ligand molar ratio has a significant effect on the performance of the material. Metal ions with good coordination ability with carboxylate groups (such as Fe(III) and Cu(II)) form solid powders that are not moldable, while those with weak coordination ability with carboxylate groups (such as Na(I)) form liquid-like gels (Supplementary Fig. 13–15). Moreover, decreasing the metal-to-ligand molar ratio from 100 to 5% decreases the glass transition temperature and makes the polymer softer (Supplementary Fig. 9 and Supplementary Table 3). All these cases are unfavorable for our study.

**Thermal healing property**. The thermal healing properties of the **PDMS-COO-Zn** polymer are illustrated in Fig. 3a, b. The samples were cut into two completely separate pieces with a razor blade. The two half-plates were brought back into contact and then healed at different temperatures for different length of time. The results show that the self-healing efficiency increases with increasing healing times and increasing healing temperature (Supplementary Fig. 16 and Supplementary Table 4). Heating to 80 °C significantly accelerates the healing process, which agrees well with the rheological and DMA analysis showed that the

modulus of the polymer decreases above this temperature. When the sample is healed at 80 °C for 4 h, the breaking strain and maximal strength are almost completely recovered compared to the original sample (Fig. 3b). The cycle of breaking and healing can be repeated many times. The notch on the film almost disappeared after healing although an indistinct healed scar was still visible under the microscope (Fig. 3c). Figure 3d shows that four pieces of sample, one of which was colored blue, can be assembled into a mini bench through thermal healing. The resulting mini bench can sustain a load of 100 g. Interestingly, the healing process can be rapid under certain conditions. We observed the healing of a crack on a successive scan timescale of 300 s in scanning electron microscopy (SEM) tests (Supplementary Fig. 17).

**Application in orthopedic immobilization**. Due to the soft and viscoelastic properties of this material at elevated temperatures, objects made from the **PDMS-COO-Zn** polymer could be easily reshaped. Figure 3e and Supplementary Movie 2 show that upon gentle heating, a fusilli can be made by twisting a dumbbell-shaped sample. After cooling, the fusilli can keep its shape and sustain a heavy load (100 g) (no apparent deformation was

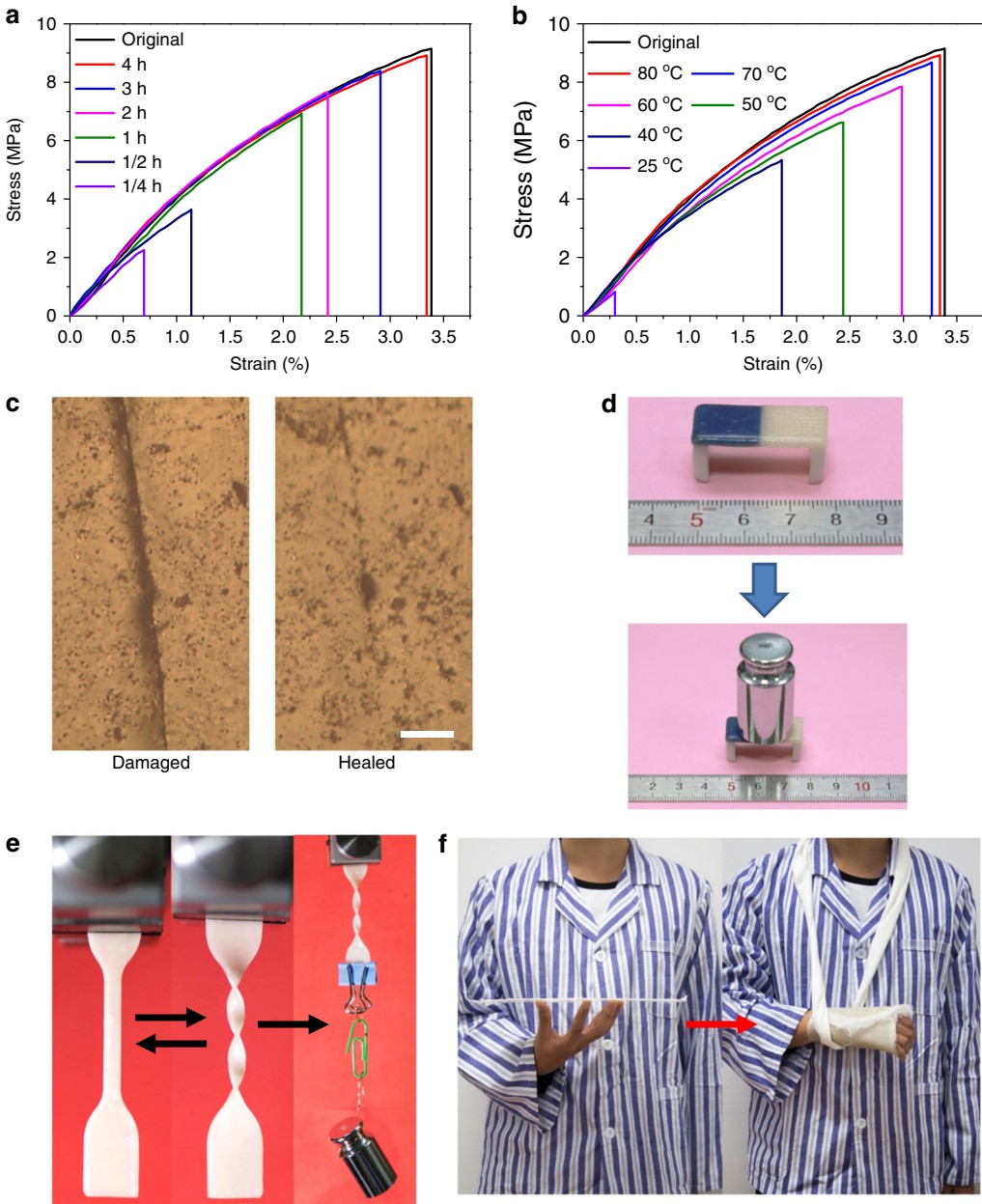

**Fig. 3** Thermal healing and reshaping properties of **PDMS-COO-Zn** polymer. **a** Flexural stress-strain curves of **PDMS-COO-Zn** before and after healing at 80 °C for different time. **b** Flexural stress-strain curves of **PDMS-COO-Zn** before and after healing at different temperature for 4 h. **c** Microscopic images of a film before (left) and after (right) healing at 80 °C for 4 h. Scale bar, 100 μm. **d** The weight-bearing test of **PDMS-COO-Zn** polymer after healing. **e** A fusilli-shaped polymer made by local heating within tens of seconds can sustain a weight of 100 g immediately. **f** A rigid flaky sample can be reshaped and adapted into an orthosis upon heating

observed within 1 h). Such a reshaping process can be accomplished by a common household blow drier within tens of seconds. Notably, compared to other malleable polymers that are based on dynamic covalent bonds, our polymer has advantages including requiring no additives (such as catalysts or solvents) and having a significantly lower reshaping temperature[52,54,57–60]. The rapid softening and hardening properties of the **PDMS-COO-Zn** polymer make it applicable in medical fields such as in orthopedic immobilization and external fixation systems. It is known that plaster-casting is needed to ensure bone healing when people suffer from bone fractures. However, plaster casting is cumbersome, inconvenient and the materials cannot be recycled after use[61]. Figure 3f and Supplementary Movie 3 show that a flaky sample can be reshaped and adapted into an orthosis upon

heating with a blow drier. The orthosis is rigid enough to restrict body movements to facilitate bone healing, yet it is lightweight, removable and recyclable.

**Application in 3D printing**. Three-dimensional (3D) printing, also known as additive manufacturing, has advanced rapidly in recent years, as it can proceed quickly, save time and money, increase design freedom, and enhance design innovation[62–65]. However, common 3D printing techniques face some challenges. One such challenge is that the whole 3D object will have to be discarded if it is broken or locally cracked by an external force. This is in contrast to traditional brickwork construction methods in which objects are assembled from independent bricks and therefore can be repaired through brick replacement when

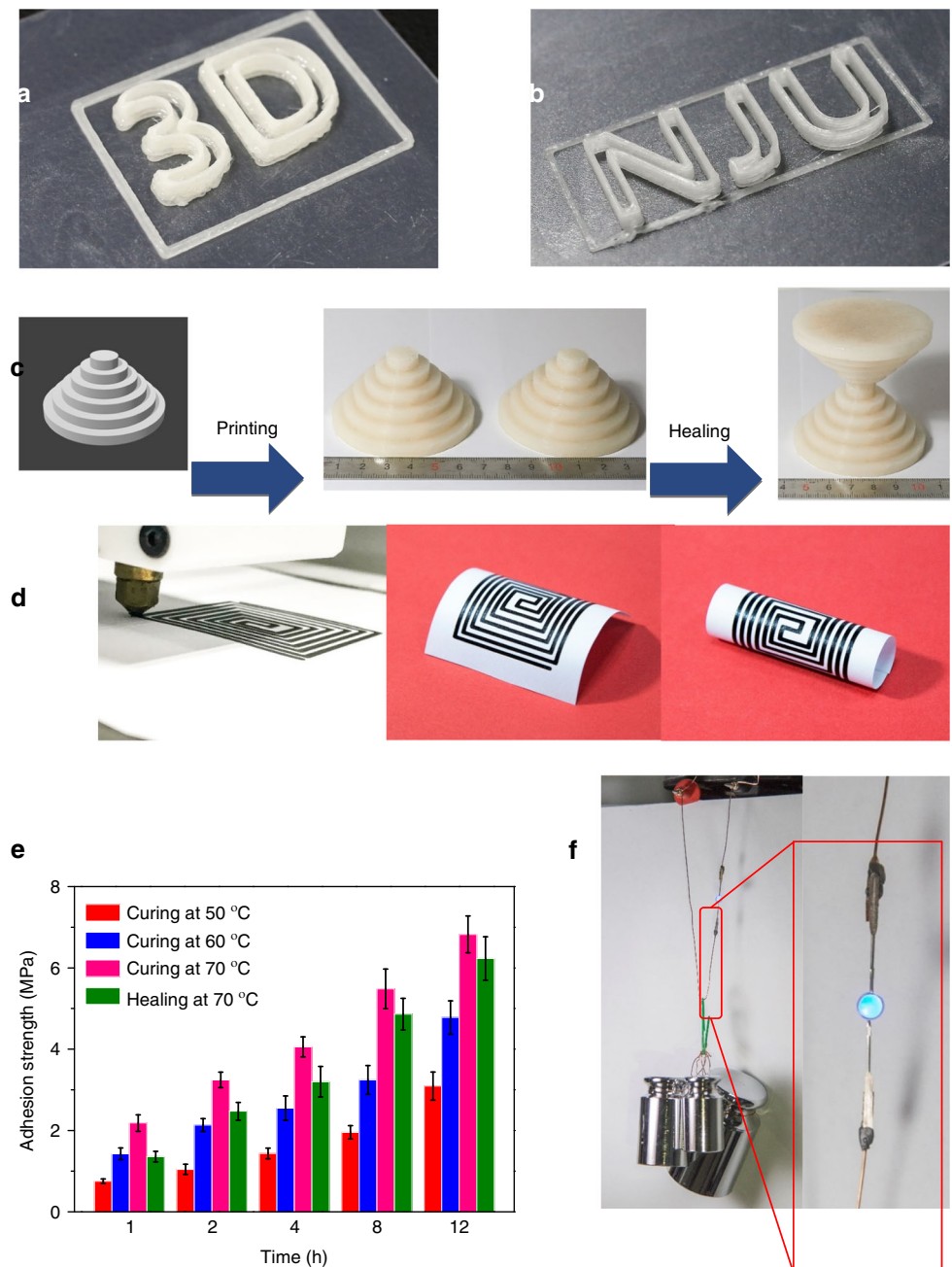

**Fig. 4** Applications of **PDMS-COO-Zn** polymer. **a**, **b** Various of 3D objects printed from the **PDMS-COO-Zn** polymer. **c** The irregular objects obtained by 3D printing. **d** An electrical circuit based on the **PDMS-COO-Zn/graphene** composite could be printed on paper, and the as-prepared device can be reshaped. **e** The effect of cure time and temperature upon lap shear adhesion with the **PDMS-COO-Zn/graphene** composite as conducting adhesive. Error bars are the s.d. from triplicate measurements. **f** A circuit that glued and conducted by **PDMS-COO-Zn/graphene** composite could sustain a weight of 800 g

damaged. Moreover, the dimensions of 3D-printed objects are restricted by the size of the printer, since a 3D printer cannot print anything bigger than itself. Our polymer is well suited to 3D printing because it turns into a viscous liquid when it is heated to 120 °C, and it quickly forms a rigid solid upon cooling. With its additional thermal healing properties, our **PDMS**-**COO-Zn** polymer can be a solution for the present problems facing 3D-printing technologies. As shown in Fig. 4a, b, Supplementary Fig. 18 and Supplementary Movie 4, we were able to print various shapes (such as 3D, NJU et. al). The 3D objects can be reconnected through thermal healing when damaged. Therefore, they do not have to be discarded if they are broken or locally cracked

by an external force. Figure 4c and Supplementary Fig. 19 show that we can construct irregular objects that are otherwise difficult to obtain. Moreover, with the self-healing feature, we can print large objects using small 3D printers. Supplementary Fig. 20 shows a wall constructed by assembling small 3D-printed bricks. After thermal-healing, the wall has excellent structural integrity and can sustain a load of 2.5 kg without obvious bending or breaking. It is anticipated that in the future, we will be able to build large houses with 3D-printed small components (Supplementary Movie 5). The small components will be connected through self-healing instead of mortar sticking. It should be noted that the healing in our system occurs via a different mechanism

than the remolding seen in common 3D-printed objects. Common 3D-printed materials can be thermally remolded to reform the 3D objects. However, their shape and function will be completely changed upon remolding. Our healing process was performed at a relative low temperature (lower than the melting point), and the shape and function do not change during the healing process (Supplementary Fig. 21).

**Application in self-healing conductors/adhesives**. By doping the polymer with conductive materials, our polymer can also be used to prepare conductive composites that are reshapable, healable, and 3D printable. Supplementary Fig. 22 shows the electric conductive properties of the composite with different doping ratios of graphene. When the content of graphene is 25 wt%, the composite has the highest electrical conduction efficiency (with the minimum dopant ratio required to obtain useful electrical conduction properties). The electric conductivity was as high as $850\,s\,cm^{-1}$. The mechanical strength of the conductive composite increases with increasing dopant ratio (Supplementary Fig. 23). The as-prepared conductive composite can be printed on various substrates and into various shapes by a common 3D printer. Figure 4d and Supplementary Movie 6 show an electrical circuit that was printed on paper. This device is rigid due to the rigidity of the conductive composite. However, it can be reshaped into various shapes such as arches and scrolls by heating and cooling, which can support themselves while functioning properly. As shown in Supplementary Fig. 24, the printed electrical circuit can power an LED. When the electrical circuit was disconnected due to local damage, it could be healed through a Joule heating effect (Supplementary Fig. 25). Paper devices of this kind would be very useful for antennas and RFID labels with enhanced durability. Moreover, our conductive composite can also function as an advanced healable conductive adhesive (Fig. 4e). When the conductive adhesive was cured at 70 °C for 12 h, the adhesion value reached a maximum of 6.8 MPa, which is much higher than that of the commercial PU-based electrically conductive adhesive (0.7 MPa, labeled 3188)[66]. When the conductive adhesive was damaged, the adhesion value can be recovered to 6.2 MPa after healing. As shown in Fig. 4f, a glued circuit can sustain a load of 800 g (with an adhesive area of approximately 10 mm²). In the past few years, much effort has been devoted to the design of easily handeled and environmentally friendly conductive adhesives to replace toxic tin-lead solder. Compared to other conductive adhesives reported to date[67–69], the **PDMS-COO-Zn/graphene** composite polymer has the advantages of being reusable and 3D printable at mild temperatures.

## Discussion

In order to overcome the incompatibility between mechanical rigidity and dynamic healing, we herein proposed a strategy for the design of rigid and healable polymers. By incorporating weak but abundant Zn(II)-carboxylate coordination interactions into a PDMS backbone, a designed polymer, that is very rigid and has a high Young's modulus (up to 480 MPa) and low elongation-at-break ( < 4%), was obtained. Increasing the temperature shifted the $Zn^{2+} + PDMS\text{-}COO^- \leftrightarrow Zn^{2+}(^-OOC\text{-}PDMS)$ equilibrium toward the dissociated state, making the polymer softer and more fluid. Therefore, the rigid **PDMS-COO-Zn** polymer became plastic, healable, reconfigurable, and re-processable upon heating. Moreover, the change in the mechanical strength upon heating or cooling is rapid and fully reversible. Based on these features, we further demonstrate that our polymer is advantageous for various applications such as orthopedic immobilization, conductive composites/adhesives, and 3D printing. We believe that the design concepts presented

here represent a general approach to the preparation of rigid and healable functional materials, and we envisage that the material reported in this work will be promising for practical applications. The properties of the **PDMS-COO-M** polymer are highly tunable by varying the content of ionic repeating units, metal-to-ligand molar ratio, and metal ions. Therefore, this system is available for further optimization based on the different requirements of various applications, and such optimizations will be the subject of future studies.

## Methods

**Materials and general measurements**. Poly(dimethylsiloxane-co-methylhydrosiloxane), trimethylsilyl terminated (PHMS, with methylhydrosiloxane 50 mol %, m = n, $M_w$ = 12000, $M_n$ = 8500, Đ = 1.4) was purchased from JOTEC New Material Technology Co., Ltd. (Hangzhou, China). Graphene (with a diameter of 5–10 μm and thickness of 3–10 nm) was commercial available from Nanjing XFNANO Materials Tech Co., Ltd. Methyl methacrylate (MMA), Karstedt catalyst solution (Pt, 2% in xylene) and other chemicals and solvents were purchased from Sigma-Aldrich. All of the chemicals were used as received without further purification. NMR (¹H and ¹³C) spectra were recorded on a Bruker DRX 400 NMR spectrometer in deuterated solvents at room temperature (25 °C). Thermogravimetric analysis (TGA) was performed on a simultaneous SDT 2960 thermal analyzer from 30 to 800 °C at a heating rate of 10 °C·min⁻¹ under a N₂ atmosphere. Differential scanning calorimetry (DSC) experiments were performed using a Mettler-Toledo DSC1 STARᵉ differential scanning calorimeter under a dry nitrogen atmosphere (50 mL min⁻¹). Temperature and enthalpy calibrations were performed before the experiments using zinc and indium standards. The temperature range was 10 °C to 140 °C, and the heating/cooling rate was 20 °C min⁻¹. The scanning electron microscopy (SEM) images were obtained on an S-4800 SEM (Hitachi, Japan) at 10 kV. FT-IR spectra were recorded with a Horiba Jobin-Yvon Fluorolog-3 fluorometer. Analytical gel permeation chromatography (GPC) were performed on a Malvern VE2001 GPC solvent/sample Module with three Visco-GELTM I-MBHMW-3078 columns. Dynamic mechanical analysis measurements were made on a dynamic mechanical analyzer (TA Instruments Q800) over temperatures ranging from −40 °C to 90 °C. The rheological behaviors were evaluated on a TA Instruments DHR-2 system. Temperature sweeps were performed with 20 mm parallel plates on circular samples 20 mm in diameter. Temperature sweeps were run from 20 °C to 120 °C (or from 120 °C to 20 °C in the cyclic test mode) at a rate of 2 °C min⁻¹ and a frequency of 1 Hz, and the strain was automatically modulated at 0.03% ± 0.02% by the instrument to keep the measured torque at a reasonable value as the sample softened. Contact with the sample was maintained by the auto-compression feature set to 0.2 ± 0.15 N. The thermal conductivities of the composites were measured using a physical property measurement system (PPMS). The electrical conductivity was measured using an Agilent 34110 A digital multimeter by the four-probe method. The adhesive tests were performed using an Instron 3343 Microtester. Lap shear testing was used to evaluate the adhesion properties, and all adhesive tests were carried out on steel substrates. The steel adherents (100 mm × 25 mm × 2 mm) were sanded with 50 grit sandpaper and then washed with soapy water and rinsed with acetone. The flaky samples used in the orthopedic immobilization were prepared from the polymer **PDMS-COO-Zn** and a general bandage by layer-by-layer hot pressing.

**Synthesis of PDMS-MMA**. The **PDMS-MMA** was synthesized by hydrosilylation reaction between PHMS and methyl methacrylate (MMA). PHMS (50 g, 0.4 mol Si-H) and MMA (48 g, 0.48 mol) were charged in a 1 L three-necked flash equipped with a reflux condenser and a magnetic stirrer. Toluene (300 mL) was added, and the mixture was stirred under an inert atmosphere to dissolve the reactant. Then, 0.5 mL of Karstedt catalyst solution (Pt, 2% in xylene) was added via a syringe, and the mixture was refluxed at 80–85 °C for 10 h. After the reaction, toluene was removed under vacuum, and the **PDMS-COOH** precursor **PDMS-MMA** was obtained as a viscous light-white liquid (89.01 g, yield 98.9%), which was used in next step without purification. Molecular weight according to GPC: $M_w$ = 27000; $M_n$ = 16000 (Đ = 1.7). ¹H NMR (400 MHz, CDCl₃, δ): 3.640 (s, 3 H), 2.593 (m, 1 H), 1.200 (d, J = 6.8 Hz, 3 H), 1.006 (m, 1 H), 0.690 (m, 1 H). As shown in Supplementary Fig. 2, the formation of **PDMS-MMA** was confirmed by the ¹H NMR spectrum; the disappearance of the peak at δ = 4.690 (the characteristic peak of Si-H) confirmed the addition reaction was complete, and the appearance of a peak at δ = 2.593 indicated that the reaction mainly affords the anti-Markovnikov product.

**Synthesis of PDMS-COOH**. The **PDMS-COOH** was synthesized by the hydrolysis of **PDMS-MMA**. The as-prepared **PDMS-MMA** (89 g, approximately 0.4 mol) was dissolved in 300 mL of THF and added to aqueous LiOH (720 mL, 1 mol L⁻¹). The mixed solution was stirred and reflux at 85 °C for 1 h. The aqueous phase was collected and washed with THF. An additional 200 mL of THF was added to the aqueous phase, and the pH was adjusted to 1~2 by 6 mol L⁻¹ aqueous hydrochloric acid solution. The THF phase was collected, and the THF was removed under

reduced pressure. The **PDMS-COOH** was obtained as a colorless and transparent viscous solid (82.04 g, yield 97.2%). Molecular weight according to GPC: $M_w = 24500$, $M_n = 13000$ ($Đ = 1.9$). $^1$H NMR (400 MHz, CDCl$_3$, δ): 11.993 (s, 1 H), 2.427 (m, 1 H), 1.110 (d, J = 7.2 Hz, 3 H), 0.948(m, 1 H), 0.620 (m, 1 H). $^{13}$C NMR (400 MHz, CDCl$_3$): δ 178.11, 25.26, 21.55, 19.79. As shown in Supplementary Fig. 3, the $^1$H NMR and $^{13}$C NMR signals at δ = 11.993 and δ = 178.11, respectively, confirmed the completion of the hydrolysis reaction of the ester and the formation of the carboxyl moieties. The band at 1707 cm$^{-1}$ in the FT-IR spectrum (Supplementary Fig.4) also confirmed the formation of the carboxyl groups.

**Synthesis of the PDMS-COO-Zn polymer**. The highly crosslinked **PDMS-COO-Zn** polymers were prepared by reacting **PDMS-COOH** with ZnCl$_2$ with the addition of Et$_3$N in ethanol. The typical procedure for the preparation of **PDMS-COO-Zn** films is as follows. ZnCl$_2$ (3.11 mL, 100 mg mL$^{-1}$) in ethanol was added to a solution of **PDMS-COOH** (1 g) in ethanol (10 mL) under stirring. Then, Et$_3$N (0.63 mL) was added into the solution dropwise. The reaction mixture was stirred for 12 h at room temperature and then concentrated. The residue was washed with dichloromethane (DCM) to afford the **PDMS-COO-Zn** polymer as a solid powder. The solid powder was then hot pressed into block samples of different dimensions for the different tests. As shown in Supplementary Fig. 4, the stretching vibration at 1709 cm$^{-1}$ of the COOH group in the **PDMS-COOH** is suppressed and two absorptions bands at 1596 cm$^{-1}$ and 1630 cm$^{-1}$ appear, which are attributable to the symmetric and asymmetric stretching vibrations of the COO$^-$ groups, respectively.

**Synthesis of PDMS-COO-Zn/graphene composite polymer**. Graphene dispersed in the ethanol was added to the pre-prepared **PDMS-COO-Zn** solution mentioned above to obtain the **PDMS-COO-Zn/graphene**solution. After ultrasonic treatment for 1 h, the resulting solution was concentrated, and the residue was washed with dichloromethane (DCM) to afford the **PDMS-COO-Zn/graphene** composite polymer as a black solid powder, which was then used for 3D printing and preparing conducting adhesives.

**ITC titration**. All titrations were performed using a Microcal ITC$_{200}$ apparatus at 298 K. The **PDMS-COOH** polymer was neutralized to **PDMS-COO-Et$_3$N** by Et$_3$N. The solutions for the titrations were prepared in anhydrous ethanol. The metal halide solution (6 mM ZnCl$_2$, 6 mM CuCl$_2$ and 4 mM FeCl$_3$) was added into the solution of **PDMS-COO-Et$_3$N** (0.6 mM) during the titrations. Blank titrations in anhydrous ethanol were performed, and the result was subtracted from the corresponding titrations to account for the effect of the dilution. The fitting was performed by using Origin software provided by Microcal.

**Mechanical and self-healing tests**. Mechanical flexural strain-stress and stress-relaxation experiments were performed using an Instron 3343 instrument under the three-point flexural mode. For all the tests, a sample size of 60 mm length × 5 mm width × 2 mm height and span of 13 mm was adopted, and three samples were tested for each polymer composition. The rate-dependent flexural strain-stress tests were carried out under ambient conditions (25 °C, 35% RH). The temperature-dependent flexural strain-stress test was carried out at a rate of 20 mm min$^{-1}$. For the self-healing tests, the film was cut into two pieces and then put together. The film was then healed at different temperatures for different durations. The healed films were then tested following the same procedure to obtain the flexural strain-stress curves. The mechanical healing efficiency, η, is defined as the ratio between the maximal strength restored relative to the original maximal strength. Values of the Young's moduli, maximal strengths, breaking strains, and healing efficiencies are presented as the means ± standard deviation according to the data from ≥ 4 trials. For measurements of the humidity-dependent mechanical properties, the sample was placed in the corresponding humidity environment for 1 day before test.

**3D Printing**. A fused deposition modeling (FDM) 3D printer (Regenovo, China) with temperature and gas pressure controllers was used in the 3D-printing experiments. The structures of the samples were designed using Pro/ENGINEER 5.0 (Parametric Technology Corporation, USA). The polymer samples were loaded into a syringe that was preheated to 130 °C for 10 min. The polymer filaments were extruded through a nozzle with a diameter of 400 µm under 0.4 MPa of air pressure to prepare the designed structure. For the construction of the wall from small 3D-printed bricks, the surfaces of the bricks were polished to ensure maximum surface contact.

**Data availability**. The data that support the findings of this study are available from the corresponding author upon reasonable request.

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

## Acknowledgements

This work was supported by the National Natural Science Foundation of China (Grant No. 21631006 and 21771100), the Natural Science Foundation of Jiangsu Province (Grant No. BK20170016 and BK20151377), the Fundamental Research Funds for the Central Universities (020514380121), and the program A for Outstanding PhD candidate of Nanjing University. We thank Prof. Jia Zhu and Miss Ning Xu for their help in the measurement of thermal conductivity, Prof. Xiao-Liang Wang for valuable discussion on the rheological measurement and Dr. Ming-Shen Lin for his help on the ITC measurement.

## Author contributions

J. C. L., C. H. L. and J. L. Z. conceived, designed and directed the project; J. C. L., L. L., D. P. W., M. H. Z., S. R. M., X. W. and K. Y. Z. performed the experiments; J. C. L., C. H. L., J. L. Z., Q. J. and X. Z. Y. analyzed the data; J. C. L., C. H. L. and J. L. Z wrote the paper. All authors discussed the results and commented on the manuscript.

## Additional information

**Competing interests:** Nanjing University has filed a patent based on this technology that names J. C. L., C. H. L., and J. L. Z. as inventors. The remaining authors declare no competing interests.

