## [Peer Review File · Nature Communications]

Reviewers' comments:

Reviewer #1 (Remarks to the Author):

the manuscript describes an interesting materials engineering experiment in which an Zn based PDMS ionomer with a very high ionic site density is used for 3D printing.

The work is more like proof of principle and lacks scientific depth or value, but could inspire the general public.

The manuscript is reasonably well written, and cites many of the relevant papers in the literature, with the exception of two papers by a German/Dutch team showing the clear link between healing kinetics and metal type and content for a related ionomer system. In these papers the correct interpretation of the rheological/ DMTA results in relation to self healing behaviour is presented. (Bose et al; Polymer, 69 (2015) 228-232 and Bose et al. Physical Chemistry Chemical Physics 16, issue 3, (2015) 1697-1704).

When expressing the performance of the material, test loads are expressed in grams and kilograms. While this is ok for the public at large, it is unscientific and loading conditions should also be translated into loading stresses using the appropriate engineering mechanics formula

In addition, in the manuscript references 47 and 31 are identical.

Finally, on page 10 line 232, the authors use the word "film". the dimensions are such that the correct expression would be "thin plate"

Reviewer #2 (Remarks to the Author):

The paper by Lai et al describes the synthesis of a metal-crosslinked PDMS polymer material and its application in 3D printing. While the idea of crosslinking PDMS polymers with metals is not new, the materials presented here do indeed display a novel combination of fluid-solid-like properties that appear to be governed largely by the metal-crosslink phase. However, in this reviewer's opinion the paper falls far short of being worthy of publication in Nature Communications primarily for the following two reasons:

- While the very solid (ceramic-like) material properties demonstrated in the paper are likely caused by the presence of the metal-crosslinked phase as proposed by the authors, the details of the metal crosslink mechanism remains very poorly characterized and highly speculative. EDS simply confirms that Zn is present in the materials and the only spectroscopic evidence is provided by FT-IR, while Raman would seem an obvious and more informative choice. Furthermore, as far as I can tell, no attempt is made at quantifying the mechanical effect of metal concentration. This is particularly surprising since this should provide an easy way to confirm whether the observed properties indeed correlate with amount of metal present in the materials as proposed. Another common test in metal-crosslinked polymer materials is to simply try another transition (or non-transition) metal to quantify to what extent the mechanical properties are indeed governed by metal-polymer interactions. Finally, I don't see any mention of what the effective metal:ligand ratio actually is in these materials?

- Without a deeper fundamental investigation of the underlying physical-chemical correlated dynamics that can help explain the observed unique material mechanics, the paper instead hinges on the application of these materials to 3D printing. In my opinion, the paper also falls short on this front. No quantitative evidence is provided to convincingly make the case that these materials indeed allow for 3D printing capacity that isn't already accessible with various types of thermo-plastic polymers. The reported "healing" temperature is already accessible with standard printable

thermo-plastics. Furthermore, labelling thermally induced adhesion/mending as "self-healing" is not correct, and this type of "healing" is simply what is already observed with standard thermo-plastic polymers near their glass transition temperature. But more important than the mislabeling of this mechanism as "self-healing", without more quantitative comparisons to already established thermally controlled 3D printing materials the case for novel 3D printable capacity is non-existent. In fact, significant research has recently been done in characterizing and understanding the critical thermal transitions at polymer-polymer interfaces during 3D printing, which is work that could/should serve as a good reference for this paper but none of this work is referenced (look up work by Kalman Migler).

In closing, I do agree that these materials could possibly possess a unique combination of solid-like properties with fast self-healing which, as the authors correctly point out, is a longstanding problem in the field of self-healing materials. Overcoming this problem would certainly represent a unique opportunity for novel 3D printing capacity (not to mention for the design of novel self-healing materials overall!). However, short of more quantitative comparisons with other thermally induced "self-healing" materials, the claim of 3D-printing (or self-healing) novelty cannot be made. Furthermore, a deeper fundamental investigation of the plausibly metal-crosslink phase controlled unique mechanical properties is warranted, and such studies could likely bring novel insights on how to design materials with kinetically sharp fluid-solid transitions (which would also be highly valuable to the field of self-healing materials). Hence, due to subpar mechanistic understanding of the presented material properties and lack of quantitative comparison to other 3D printable (or self-healing) polymer materials, this reviewer cannot recommend publication in Nature Communications.

Reviewer #3 (Remarks to the Author):

In this manuscript, Li and coworkers reported the fabrication of a thermoplastic ionomer (PDMS-COO-Zn) by crosslinking carboxylate group-containing polydimethylsiloxane polymers (PDMS-COOH) with zinc ions. The electrostatic and coordination bonds in the PDMS-COO-Zn ionomers are highly dynamic above 80 °C, endowing the ionomers with high flowability under improved temperature. Therefore, the PDMS-COO-Zn ionomers can be used to fabricate 3D objects via 3D printing. Moreover, benefiting from their thermoplasticity, these 3D objects are able to heal mechanical damages such as cuts and cracks at 80 °C. The self-healing function and mechanical properties are interesting. However, the synthesis of PDMS-COO-Zn, which was claimed as a new polymer by the authors, has been reported by Cohen and coworkers in 2006. In their papers (Polymer 46, 2005, 12416–12421; Macromolecules 2006, 39,426-438; Macromolecules, 2006, 39, 2398-2404), synthesis, rheology and mechanical properties of the PDMS-COO-Zn have already been investigated in detail. Even though Cohen and coworkers did not report that the PDMS-COO-Zn has healing ability and can be used to print 3D objects, these results are quite obvious when considering the thermoplasticity of PDMS-COO-Zn. Therefore, I consider this paper is lack of novelty and should be published on a more specialized journal.

Response to the referee:

Reviewer #1 (Remarks to the Author):

The manuscript describes an interesting materials engineering experiment in which an Zn based PDMS ionomer with a very high ionic site density is used for 3D printing.

Response: We thank the reviewer for the positive comments.

The work is more like proof of principle and lacks scientific depth or value, but could inspire the general public.

Response: We thank the reviewer for his/her comments. As pointed by the third reviewer, our material possesses a unique combination of solid-like properties with fast self-healing which is a longstanding problem in the field of self-healing materials. Due to the slow diffusion dynamics, it is highly challenging to achieve self-healing in rigid and solid-like materials. It is noted that, very recently, Prof. Takuzo Aida et al reported that a polymer cross-linked by dense hydrogen bonds can be both mechanically robust and repairable (*Science*, DOI: 10.1126/science.aam7588). Therefore, we believe this work not only can inspire the general public but also has important scientific significance.

The manuscript is reasonably well written, and cites many of the relevant papers in the literature, with the exception of two papers by a German/Dutch team showing the clear link between healing kinetics and metal type and content for a related ionomer system. In these papers the correct interpretation of the rheological/ DMTA results in relation to self-healing behaviour is presented. (Bose et al; *Polymer*, 69 (2015) 228-232 and Bose et al. *Physical Chemistry Chemical Physics* 16, issue 3, (2015) 1697-1704).

Response: We thank the reviewer for reminding us the references by German/Dutch team. They have conducted excellent work on the correlation between network dynamics and the macroscopic surface scratch healing behavior for a series of elastomeric ionomers. According to their method, we have obtained the temperature dependent relaxation timescale (τ) of our polymer from frequency sweeps in oscillatory shear rheology (Figure C1). The results showed that the relaxation time is decreasing with increasing of temperature, thus validating the better thermal-healing behavior at high temperatures. However, a directly correlation between the polymer chain relaxation timescale (τ) and macroscopic healing timescale (t_{heal}) is difficult for our system. The relaxation timescale (τ) of our polymer is about 47.8 s at 25 °C, indicating that the polymer can even be healed at room temperature according to the literature by Bose et al. However, this is not true in our experiments. Actually, the recent paper by Aida et al. also observed a contradiction between the relaxation time and healing behavior (*Science*, DOI: 10.1126/science.aam7588).

Figure C1. DSC Frequency sweep measurement of PDMS-COO-Zn at different temperature.

When expressing the performance of the material, test loads are expressed in grams and kilograms. While this is ok for the public at large, it is unscientific and loading conditions should also be translated into loading stresses using the appropriate engineering mechanics formula.

Response: The mechanical strength was investigated by flexural testing from which the Young's modulus can be determined by the formula $E = (F \times L)/(A \times \Delta L)$. The image for weight-bearing test is used only for demonstrating that our material can sustain a heavy load without obvious bending or breaking. Such presentations are common in the literature (Huang et al, Nat. Commun., 6 (2015), 10310; Sun et al, Nat. Mater., 12 (2013), 932-937).

In addition, in the manuscript references 47 and 31 are identical.

Response: We are sorry for the duplication of references 47 and 31. We have removed one of them.

Finally, on page 10 line 232, the authors use the word "film". The dimensions are such that the correct expression would be "thin plate"

Response: The word "film" was replaced by "thin plate" in the relating sentence.

Reviewer #2 (Remarks to the Author):

The paper by Lai et al describes the synthesis of a metal-crosslinked PDMS polymer material and its application in 3D printing. While the idea of crosslinking PDMS polymers with metals

is not new, the materials presented here do indeed display a novel combination of fluid-solid-like properties that appear to be governed largely by the metal-crosslink phase.

Response: We thank the reviewer for the positive comments on the novel property of our material.

However, in this reviewer's opinion the paper falls far short of being worthy of publication in Nature Communications primarily for the following two reasons:

- While the very solid (ceramic-like) material properties demonstrated in the paper are likely caused by the presence of the metal-crosslinked phase as proposed by the authors, the details of the metal crosslink mechanism remains very poorly characterized and highly speculative. EDS simply confirms that Zn is present in the materials and the only spectroscopic evidence is provided by FT-IR, while Raman would seem an obvious and more informative choice. Furthermore, as far as I can tell, no attempt is made at quantifying the mechanical effect of metal concentration. This is particularly surprising since this should provide an easy way to confirm whether the observed properties indeed correlate with amount of metal present in the materials as proposed. Another common test in metal-crosslinked polymer materials is to simply try another transition (or non-transition) metal to quantify to what extent the mechanical properties are indeed governed by metal-polymer interactions. Finally, I don't see any mention of what the effective metal:ligand ratio actually is in these materials?

Response: According to the reviewer's suggestion, we have performed a series of control experiments to investigate the relationships between the mechanical properties and the amount/type of metal-crosslinks. Firstly, we measured the association constant (K_a) from

isothermal calorimetric titrations (ITC) studies. The Zn(II)-carboxylate association constant was about $4.10 \times 10^4 \text{ M}^{-1}$, which is quite small compared to typical coordination bonds, but is still much higher than hydrogen bonding. The Raman spectra can not be obtained due to the emission from PDMS-COO-Zn polymer, which is also an indication that coordination bonds were formed. Secondly, we studied the effect of neutralization level on the properties of the resulting materials. The results show that the T_g and Young's moduli of the polymer increase with the increase of neutralization level (Figure C2 and Table C1). Therefore, we adopted 100% of neutralization level (Corresponding to metal-to-ligand molar ratio of 2:1) which provide samples with best performance.

Figure C2. DSC curves of the PDMS-COO-Zn polymer with different neutralized ratio, the yellow square shows the T_g of corresponding curves, which revealed the T_g was downshift with decreasing the neutralized ratio and the T_g of 100% neutralized was at about 55.7 °C.

Table C1. Neutralized Ratio dependence of T_g and Young's modulus of **PDMS-COO-Zn**.

Neutralized Ratio	Metal to Ligand Molar Ratio	T_g ($^{\circ}\text{C}$)	Young's Modulus (MPa)
100%	1:2	55.7	478.13
75%	3:8	38.4	285.27
50%	1:4	27.2	112.15
25%	1:8	2.5	12.96
10%	1:20	-6.8	1.58
5%	1:40	-14.6	0.25

Thirdly, we studied the effect of different metal ions. We found metal ions with good coordination ability toward carboxylate groups (such as Fe(III) and Cu(II)) form unmoldable solid powder, while those with weak coordination ability toward carboxylate groups (such as Na(I)) form liquid-like gels, both of which are unfavorable for our study (Figure C3-C5). These results indicate that Zn(II)-carboxylate crosslinks with moderate strength facilitate the fabrication of rigid and healable polymer.

Figure C3. Optical Photograph of **PDMS-COO-M** ($M=\text{Fe}^{3+}$, Cu^{2+} , Na^+), which shows that the **PDMS-COO-Fe** and **PDMS-COO-Cu** was unmoldable solid powder and **PDMS-COO-Na** was liquid-like gels.

Figure C4. The ITC titration data of the **PDMS-COO⁻** with FeCl_3 in anhydrous ethanol at 25 °C, which shows the K_a of **PDMS-COO-Fe** was about 8 times as much as **PDMS-COO-Zn**.

Figure C5. The ITC titration data of the **PDMS-COO⁻** with **CuCl₂** in anhydrous ethanol at 25 °C, which shows the K_a of **PDMS-COO-Cu** was about 28 times as much as **PDMS-COO-Zn**.

Based on these new results, we proposed a different mechanism for the novel property of our polymer. The coordination bonds used in our study is weak but still significantly stronger than hydrogen bonding, therefore the resulting polymer is very strong and rigid at room temperature. Due to the weak coordination bond energy, the coordination equilibrium is sensitive to temperature. When heated, the $Zn^{2+} + PDMS-COO^- \leftrightarrow Zn^{2+}(-OOC-PDMS)$ equilibrium was shifted toward the disassociated state so that non-crosslinked **PDMS-COO⁻** chains was generated progressively, which reduces the cross-linking density and enhance the polymer chain mobility. The mechanical strength of the polymer is modulated by the cross-linking density. Therefore, the higher the temperature, the more non-crosslinked **PDMS-COO⁻** chains was generated, and therefore the polymer became more and more soft and viscoelastic. Upon cooling, the $Zn^{2+} + PDMS-COO^- \leftrightarrow Zn^{2+}(-OOC-PDMS)$ equilibrium was shifted toward the associated state, leading to the restoration of the

densely crosslinked three dimensional network. Such reversible processes make the mechanical properties of PDMS-COO-Zn polymer highly sensitive to temperature, which are favorable for healing, reshaping upon heating (Figure C6).

Figure C6. Schematic structure of the polymer network at different temperatures.

- Without a deeper fundamental investigation of the underlying physical-chemical correlated dynamics that can help explain the observed unique material mechanics, the paper instead hinges on the application of these materials to 3D printing. In my opinion, the paper also falls short on this front. No quantitative evidence is provided to convincingly make the case that these materials indeed allow for 3D printing capacity that isn't already accessible with various types of thermo-plastic polymers. The reported "healing" temperature is already accessible with standard printable thermo-plastics. Furthermore, labelling thermally induced adhesion/mending as "self-healing" is not correct, and this type of "healing" is simply what is already observed with standard thermo-plastic polymers near their glass transition temperature. But more important than the mislabeling of this mechanism as "self-healing", without more quantitative comparisons to already established thermally controlled 3D printing materials the case for novel 3D printable capacity is non-

existent. In fact, significant research has recently been done in characterizing and understanding the critical thermal transitions at polymer-polymer interfaces during 3D printing, which is work that could/should serve as a good reference for this paper but none of this work is referenced (look up work by Kalman Migler).

Response: We thank the reviewer for his/her comments. Generally, a polymer can be used for 3-D printing if it turns into viscous liquid when being heated and quickly forms rigid solid upon cooling. Therefore, most thermal conductive thermo-plastic polymers can be used for 3-D printing. The most interesting feature of our material for application in 3-D printing is that objects printed by our polymer can be healed when damaged. The healing in our system is different from remolding of common of 3D objects. Common 3D printing materials can be thermally remolded to reform the 3D objects. However, their shape and function will be totally changed upon remolding. Our healing process was performed at relative low temperature (lower than the melting point), and the shape and function didn't change during the healing process. We agree with the reviewer that the word "self-healing" is not appropriate for describing the properties of our material as external stimuli (heating) is need for healing. Therefore, we use "thermal healing" in the revised manuscript. The paper by Kalman Migler was cited in the revised manuscript.

In closing, I do agree that these materials could possibly possess a unique combination of solid-like properties with fast self-healing which, as the authors correctly point out, is a longstanding problem in the field of self-healing materials. Overcoming this problem would certainly represent a unique opportunity for novel 3D printing capacity (not to mention for the design of novel self-healing materials overall!). However, short of more quantitative comparisons with other thermally induced "self-healing" materials, the claim of 3D-printing

(or self-healing) novelty cannot be made. Furthermore, a deeper fundamental investigation of the plausibly metal-crosslink phase controlled unique mechanical properties is warranted, and such studies could likely bring novel insights on how to design materials with kinetically sharp fluid-solid transitions (which would also be highly valuable to the field of self-healing materials).

Response: As acknowledged by the reviewer, our material possesses a unique combination of solid-like properties with fast self-healing which is a longstanding problem in the field of self-healing materials. In the revised manuscript, we have added more quantitative comparisons with other thermally healable materials and performed more control experiments to understand the structure-properties relationships. The sharp fluid-solid transition is very unique as compared other materials and would also be highly valuable to the field of self-healing materials

Hence, due to subpar mechanistic understanding of the presented material properties and lack of quantitative comparison to other 3D printable (or self-healing) polymer materials, this reviewer cannot recommend publication in Nature Communications.

Response: According to the reviewer's comments, we have carefully revised the manuscript. We believe that the manuscript has been significantly improved and should be suitable for publication now.

Reviewer #3 (Remarks to the Author):

In this manuscript, Li and coworkers reported the fabrication of a thermoplastic ionomer (PDMS-COO-Zn) by crosslinking carboxylate group-containing polydimethylsiloxane polymers (PDMS-COOH) with zinc ions. The electrostatic and coordination bonds in the PDMS-COO-Zn ionomers are highly dynamic above 80 °C, endowing the ionomers with high flowability under improved temperature. Therefore, the PDMS-COO-Zn ionomers can be used to fabricate 3D objects via 3D printing. Moreover, benefiting from their thermoplasticity, these 3D objects are able to heal mechanical damages such as cuts and cracks at 80 °C. The self-healing function and mechanical properties are interesting. However, the synthesis of PDMS-COO-Zn, which was claimed as a new polymer by the authors, has been reported by Cohen and coworkers in 2006. In their papers (Polymer 46, 2005, 12416–12421; Macromolecules 2006, 39,426-438; Macromolecules, 2006, 39, 2398-2404), synthesis, rheology and mechanical properties of the PDMS-COO-Zn have already been investigated in detail. Even though Cohen and coworkers did not report that the PDMS-COO-Zn has healing ability and can be used to print 3D objects, these results are quite obvious when considering the thermoplasticity of PDMS-COO-Zn. Therefore, I consider this paper is lack of novelty and should be published on a more specialized journal.

Response: We thank the reviewer for his/her constructive comments. In our previous submission, we categorized our material as an ionomer in our previous manuscript considering that it contains both neutral repeating units ($-\text{[Si(CH}_3)_2]_n-$) and ionized repeating units ($-\text{[Si(CH}_3)(\text{R-COO}^-)]_n-$) while the linear oligomer was thought to be crosslinked through $\text{Zn}^{2+} - \text{R-COO}^-$ ionic interactions. Based on this assignment, the our polymer is not new since Cohen and coworkers have published several paper on the structures and properties ionomers formed between Zn(II) and poly (ethylene-co-methacrylic acid). However, in our revision, both small-angle X-ray scattering (SAXS) analysis and energy dispersive X-ray spectroscopy (EDS) pictures did not reveal the typical aggregations for ionomers. The

characteristic order-to-disorder transition (T_i) of ionic clusters disappeared when we removed the triethylamine hydrochloride impurity (which does not influence the mechanical and self-healing properties of our materials and therefore we didn't remove it before). The IR spectroscopy revealed clearly stretching peaks characteristic for formation of coordination bonds instead of ionic interactions. Moreover, according to the IUPAC definition (Pure and Applied Chemistry, 1996, 68(12): 2287–2311), an ionomer is a polymer that comprises repeat units of both electrically neutral repeating units and a fraction of ionized units (usually no more than 15 mole percent), but our polymer contains 50 mole percent of ionized units. Therefore, our material should be significantly different from the ionomers reported by Cohen and coworkers. Our material is a polymer crosslinked by weak but abundant coordination bonds, which is unprecedented in the literatures. It is true that the ionomer reported by Cohen and coworkers will have self-healing. However, the mechanical strength of our materials is several orders higher than those ionomers. Achieving self-healing (even thermal healing) property in such materials is highly challenging due to the reduced polymer chain mobility and slow diffusion dynamics.

Reviewers' comments:

Reviewer #1 was unable at this time to look over the authors' response to their comments. Therefore, reviewer #3 looked over the authors' response and thought it was satisfactory.

Reviewer #2 (Remarks to the Author):

The revised paper is significantly improved and could potentially be acceptable for publication in Nature Communications. However, the following issues still needs to be addressed:

- In line 131-133 the authors claim that the reported thermally controlled change in strength has 'never been reported'. Are the authors sure about the validity of this statement? Does this apply to all classes of materials including metals and ceramics? Unless the authors can verify the uniqueness of these properties, it is strongly recommended to tone down the value of this statement.

- According to Fig. S9 the bulk relaxation time of the material is on the order of tens of seconds at 25C. If this is true, then the material should not be able to keep it's shape over very long time scales at room temperature. Is this true? If so, the authors clearly need to point this out, since the applicability for 3D printing functional load-bearing objects for example, would be severely limited. If this is not the case (i.e. if the material can indeed hold its shape over much longer periods of time), then I don't see how the frequency sweep data in Fig. S9 can be explained?

- Why is the viscoelastic fluid-solid transition temperature different when measured in shear vs compression (Fig. 2c vs 2e)? Please provide an explanation.

- I have never seen the term "neutralized ratio" used before in the context of metal:ligand ratio. Can the authors please explain what is meant by this? If they simply mean to report the amount of metal relative to ligand, then why not just use the metal:ligand ratio, which seems to be the standard figure of merit in this context in metal-coordinate polymer materials studies?

- In line 196-197 the authors state that the material can keep an induced twisted shape under heavy load without deformation. Again, going back to the point above regarding rheologically observed bulk relaxation time, this is surprising. How long can this shape be held without significant creep?

Reviewer #3 (Remarks to the Author):

The fabrication of mechanically strong and healable polymers is challenging because the mobility of polymer chains is restricted and the healing process is difficult to proceed. The present manuscript presents the fabrication of rigid and healable PDMS-COO-Zn polymers by exploiting the reversibility of Zn(II)-carboxylate interactions. Although the coordination interactions of Zn(II)-carboxylate are relatively weak, an abundant number of Zn(II)-carboxylate interactions on PDMS side chains can produce polymers with significantly enhanced mechanical properties. Upon heating, the PDMS-COO-Zn polymers exhibit satisfactory self-healing capacity. Moreover, the soft-rigid transition property makes the PDMS-COO-Zn polymers suitable for 3D printing. The application of PDMS-COO-Zn in orthopedic immobilization was also demonstrated. The method developed in this work is useful for the design of other kinds of healable polymer composites with enhanced mechanical robustness. The revised manuscript, which reasonably addressed the questions raised by the previous reviewers, was well-written, and is recommended for publication in Nature Communication after minor revisions shown below:

(i) When PDMS-COO-Zn polymers were doped with graphene (for example, 25 wt% graphene), the conductivity of the polymer composites should be given in the main text. Moreover, the mechanical properties of graphene/PDMS-COO-Zn should be characterized.

(ii) Films should be replaced with "thin plates".

Point-by-point response to reviewers' comments:

Reviewer #2 (Remarks to the Author):

The revised paper is significantly improved and could potentially be acceptable for publication in Nature Communications. However, the following issues still needs to be addressed:

Response: We appreciate your positive comments.

- In line 131-133 the authors claim that the reported thermally controlled change in strength has 'never been reported'. Are the authors sure about the validity of this statement? Does this apply to all classes of materials including metals and ceramics? Unless the authors can verify the uniqueness of these properties, it is strongly recommended to tone down the value of this statement.

Response: Thank you for your suggestion. We agree that it is not appropriate to claim that a thermally controlled change in strength has never been reported. We have toned down the statement by revising this sentence to “Such a significant change in the mechanical strength under a narrow temperature range has never been reported for self-healing polymers”.

- According to Fig. S9 the bulk relaxation time of the material is on the order of tens of seconds at 25C. If this is true, then the material should not be able to keep it's shape over very long time scales at room temperature. Is this true? If so, the authors clearly need to point this out, since the applicability for 3D printing functional load-bearing objects for example, would be severely limited. If this is not the case (i.e. if the material can indeed hold its shape over much longer periods of time), then I don't see how the frequency sweep data in Fig. S9 can be explained?

Response: Thank you for your valuable comment, which reminded us of the inconsistency between the material's performance and the relaxation time revealed by the rheological data. During the revision of this manuscript, we re-measured the rheological properties of our polymer with both old samples and fresh samples. The new frequency sweep curves at 25°C

(35% RH) did not show a cross point between G' and G'' even when the frequency was reduced to 10^{-2} rad/s (**Figure R1**). After many control experiments, it was found that the observation of a cross point between G' and G'' in our original data was due to the high humidity in our test. As water is also a weakly coordinating ligand for Zn(II) ions, some carboxylate ligands will be replaced by water molecules in high humidity environments, which reduces the crosslinking density and therefore weakens the mechanical properties. As shown in **Figure R2**, both the Young's modulus and the relaxation time of the polymer decreased with increasing environmental humidity (the measured sample was placed in the corresponding humidity environment for 1 day before testing). Interestingly, this process is reversible. The polymer regains its strength and rigidity when the humidity is decreased. We performed a long-term (one month, three samples) monitoring of the Shore-D hardness of our polymer with the variance of environmental humidity (**Figure R3**). The results showed that the Shore-D hardness of the samples decreased with the increase of humidity but will be recovered when the humidity decreases. This phenomenon is reasonable since once the coordinated water molecules have been removed, the Zn(II) ions coordinate to the carboxylates again, and the crosslinking density of the polymer increases and the material returns to its rigid state.

Based on these observations, we can claim that the 3D-printed objects are stable under most environmental conditions. As shown in **Figure R4**, the 3D-printed objects demonstrated in our original manuscript still hold their shape after approximately 14 months (the pictures in **Figure R4a** and **R4c** were taken on January 4, 2017, and the pictures in **Figure R4b** and **R4d** were taken on March 23, 2018. There are no obvious changes in shape after approximately 14 months). However, prolonged load-bearing for the 3D printing objects would be difficult. As revealed by weight bearing test (**Figure R5**), our material will gradually deform upon load-bearing. This is reasonable because, unlike the permanent networks constructed by covalent bonds, our polymer cross-linked by dynamic coordination bonds must have a bulk relaxation process although the relaxation time cannot be determined in our experiment.

In the revised manuscript, the incorrect data in Supplementary **Figure 9** in our original manuscript was replaced with the corrected data as shown in **Figure R1**. The results of frequency sweep measurement of PDMS-COO-Zn at different relative humidity (**Figure R2**) and the relationship between relative humidity and Shore hardness of sample (**Figure R3**) has been added (Supplementary **Figure 10 and 11**). Moreover, considering the sensitivity of materials properties to humidity, we have renewed the data of rheological and

dynamic mechanical analysis with those obtained at fixed humidity (35% RH). The rheological data shows that the storage modulus is 470 MPa at 25°C and 0.06 MPa at 125 °C, and the soft–rigid switching ability (G'_{\max}/G'_{\min}) is approximately 8000 in the revised manuscript. These changes will not influence the content, framework as well as the conclusions of the paper.

Figure R1. Frequency sweep measurement of PDMS-COO-Zn at different temperature.

Figure R2. Frequency sweep measurement of PDMS-COO-Zn at different relative humidity. The bright green squares represent the intersection of G' and G'' .

Figure R3. The relative humidity and Shore hardness of sample for 1 month.

Figure R4. Comparison photos of 3D printed samples at intervals of 14 months. a) and c) were taken on January 4, 2017; b) and d) were taken on March 23, 2018.

Figure R5. The weight bearing tests under the single cantilever mode.

- Why is the viscoelastic fluid-solid transition temperature different when measured in shear vs compression (Fig. 2c vs 2e)? Please provide an explanation.

Response: Thank you. We have repeated the rheology measurements with fresh samples in 35% relative humidity, and the new results show that the viscoelastic fluid-solid transition temperatures are consistent with each other, and the curves formerly shown in Fig. 2c, 2d and 2e were replaced by the revised curves shown in **Figures R6, R7 and R8**.

Figure R6. The renewed temperature dependent rheological measurements of PDMS-COO-Zn polymer.

Figure R7. The renewed cyclic temperature-sweep rheological measurements of PDMS-COO-Zn polymer.

Figure R8. The renewed temperature-sweep dynamic mechanical analysis result of **PDMS-COO-Zn** polymer.

- I have never seen the term "neutralized ratio" used before in the context of metal:ligand ratio. Can the authors please explain what is meant by this? If they simply mean to report the amount of metal relative to ligand, then why not just use the metal:ligand ratio, which seems to be the standard figure of merit in this context in metal-coordinate polymer materials studies?

Response: Thank you. The "neutralized ratio" means the "metal-to-ligand molar ratio". We have changed the term "neutralized ratio" to "metal-to-ligand molar ratio" in our revised manuscript.

- In line 196-197 the authors state that the material can keep an induced twisted shape under heavy load without deformation. Again, going back to the point above regarding rheologically observed bulk relaxation time, this is surprising. How long can this shape be held without significant creep?

Response: Thank you. As mentioned above, the revised frequency sweep curves at 25 °C did not show the cross point of G' and G'' even when the frequency was reduced to 10^{-2} rad/s, indicating that the material should be able to keep its shape over relatively long timescales at room temperature. The twisted object had difficulty retaining its shape under a heavy load (more than 50-times to its own weight) without deformation since the material is still not strong enough to withstand a large force (**Figure R9**). However, if a smaller load (approximately equal to its own weight) was applied, the twisted object can keep its shape without significant creep for at least one month (**Figure R10**).

Figure R9. The heavy weight bearing test of the twisted sample, which shows that the heavy load can lead significant creep in two days.

Figure R10. The light weight bearing test of the twisted sample, which shows that the light load only can lead slight creep even in one month.

Reviewer #3 (Remarks to the Author):

The fabrication of mechanically strong and healable polymers is challenging because the mobility of polymer chains is restricted and the healing process is difficult to proceed. The present manuscript presents the fabrication of rigid and healable PDMS-COO-Zn polymers by exploiting the reversibility of Zn(II)-carboxylate interactions. Although the coordination interactions of Zn(II)-carboxylate are relatively weak, an abundant number of Zn(II)-carboxylate interactions on PDMS side chains can produce polymers with significantly enhanced mechanical properties. Upon heating, the PDMS-COO-Zn polymers exhibit satisfactory self-healing capacity. Moreover, the soft-rigid transition property makes the PDMS-COO-Zn polymers suitable for 3D printing. The application of PDMS-COO-Zn in orthopedic immobilization was also demonstrated. The method developed in this work is useful for the design of other kinds of healable polymer composites with enhanced

mechanical robustness. The revised manuscript, which reasonably addressed the questions raised by the previous reviewers, was well-written, and is recommended for publication in Nature Communication after minor revisions shown below:

Response: Thank you for your positive comments.

(i) When PDMS-COO-Zn polymers were doped with graphene (for example, 25 wt% graphene), the conductivity of the polymer composites should be given in the main text. Moreover, the mechanical properties of graphene/PDMS-COO-Zn should be characterized.

Response: Thank you. The conductivity of the polymer composites with 25 wt% graphene has been added to the main text. The mechanical properties of graphene/PDMS-COO-Zn composites with different ratios of the materials have been characterized and are shown in **Figure R11** and Supplementary **Figure 22**.

Figure R11. The Three-point flexural stress-strain curves of the conducting composites with different doping ratio.

(ii) Films should be replaced with “thin plates”.

Response: Thank you for pointing this out. We have replaced “films” with “thin plates” in the revised manuscript.

REVIEWERS' COMMENTS:

Reviewer #2 (Remarks to the Author):

The revised manuscript is improved enough to justify publication as is.

Reviewer #3 (Remarks to the Author):

I am satisfied with the revised manuscript and recommend its publication in Nature Communications without further modification.

Manuscript ID: NCOMMS-17-06141C

Title: A Rigid and Healable Polymer Cross-linked by Weak but Abundant Zn(II)-Carboxylate Interactions

Reviewer #2 (Remarks to the Author):

The revised manuscript is improved enough to justify publication as is.

Response: Thank you for your positive comments.

Reviewer #3 (Remarks to the Author):

I am satisfied with the revised manuscript and recommend its publication in Nature Communications without further modification.

Response: Thank you for your positive comments.